# Comprehensive Analysis of Learning Cases in an Autonomous Navigation Task for the Evolution of General Controllers

**Enrique Naredo** [1,*], **Candelaria Sansores** [1], **Flaviano Godinez** [2], **Francisco López** [1], **Paulo Urbano** [3], **Leonardo Trujillo** [4,5] and **Conor Ryan** [6]

1. Departamento de Ciencias Básicas e Ingenierías, Universidad del Caribe, Esquina Fraccionamiento, Tabachines, Cancún 77528, Mexico
2. Facultad de Matemáticas, Universidad Autónoma de Guerrero, Chilpancingo de los Bravo 39087, Mexico
3. Department of Computer Science, Faculty of Science, Universidade Nova de Lisboa, 2780-157 Oeiras, Portugal
4. Departamento de Ingeniería Eléctrica Electrónica, Posgrado en Ciencias de la Ingeniería, Tecnológico Nacional de México/IT de Tijuana, Tijuana 22414, Mexico
5. LASIGE, Faculty of Sciences, University of Lisbon, 1749-016 Lisbon, Portugal
6. Department of Computer Science and Information Systems, University of Limerick, V94 T9PX Limerick, Ireland
* Correspondence: enaredo@ucaribe.edu.mx

**Abstract:** Robotics technology has made significant advancements in various fields in industry and society. It is clear how robotics has transformed manufacturing processes and increased productivity. Additionally, navigation robotics has also been impacted by these advancements, with investors now investing in autonomous transportation for both public and private use. This research aims to explore how training scenarios affect the learning process for autonomous navigation tasks. The primary objective is to address whether the initial conditions (learning cases) have a positive or negative impact on the ability to develop general controllers. By examining this research question, the study seeks to provide insights into how to optimize the training process for autonomous navigation tasks, ultimately improving the quality of the controllers that are developed. Through this investigation, the study aims to contribute to the broader goal of advancing the field of autonomous navigation and developing more sophisticated and effective autonomous systems. Specifically, we conducted a comprehensive analysis of a particular navigation environment using evolutionary computing to develop controllers for a robot starting from different locations and aiming to reach a specific target. The final controller was then tested on a large number of unseen test cases. Experimental results provide strong evidence that the initial selection of the learning cases plays a role in evolving general controllers. This work includes a preliminary analysis of a specific set of small learning cases chosen manually, provides an in-depth analysis of learning cases in a particular navigation task, and develops a tool that shows the impact of the selected learning cases on the overall behavior of a robot's controller.

**Keywords:** navigation robotics; generalization; grammatical evolution

## 1. Introduction

Currently, robotics influences our modern life at work and, more recently, also in our home, as robotics has become a key technology that has created a wide range of autonomous devices that interact with their environment, transforming our lives and work practices [1,2]. An important niche for robotics is the industrial sector, where companies invest a lot in robotics and automation. For example, the [3] report from the International Federation of Robotics (IFR) shows how China has now overtaken the United States for fifth place in investment in robotics. However, the Republic of Korea remains the champion, with more than 1000 industrial robots per 10,000 employees, as reported in 2021. More recently,

we can notice how autonomous robots, being part of an emerging industry, have changed transportation as we know it to make way for new technologies such as self-driving cars and automated guided vehicles (AGVs) with the promise of improving efficiency and safety, increasing autonomous navigation as an essential issue in the field of robotics [4].

A typical navigation task is to reach a goal from a certain location. This problem has already been solved using a wide set of different methods to get a navigation controller that solves this problem [5]. A more interesting task is the design or creation of a controller that has generalization capacity, that is, the controller allows the robot to find the target from a wide set of positions. The creation of controllers can be done by an expert in the area or with methods that do it automatically. We opted for the option of automating the process and, in this case, evolutionary algorithms have proven to be a great tool. Therefore, in this work we use grammatical evolution (GE).

GE is a machine learning (ML) tool that typically uses reinforcement learning (RL) to evolve navigation controllers in robotics. It is important to highlight a well known issue related to ML models: they highly depend on the training data. The performance of ML algorithms is strongly influenced by the quality of the data used for training. Poor data can impede the model's ability to identify important features and relationships needed to make accurate predictions. In particular, training data are the most crucial elements in ML and artificial intelligence. More specifically, we treat the initial conditions used to evolve controllers as the training set. The conventional initial conditions used in a navigation task are the initial position and initial orientation of the robot.

In this article, we are interested in studying how the training or learning cases in an autonomous navigation task influence the learning process to produce controllers that can solve the same task from a set of unseen (test) cases, that is, controllers with generalization capabilities. Generalization is a key issue in ML applications where the goal is to develop solutions that can be generalized to several different scenarios rather than those optimized for a particular problem instance. An example of a generalization problem is trying to reach the same goal from a set of new cases in the same navigation environment. This is not a trivial problem and requires the development of generalization capabilities in the controller. In this sense, the training cases are the set of cases used to evolve a controller, and test cases are the set of cases to test the controller.

Our main purpose is to perform a comprehensive analysis of the entire set of cases of a particular navigation environment, placing a robot at each position considering four orientations as the learning cases in the environment to get a controller through evolutionary computation to reach a given target and then testing the controller with a hundred of different unseen cases. In this sense, the closest studies are related to route planning for autonomous mobile robots; for example, Sánchez et al. [5] presents an extensive review, including evolution-based routing algorithms; however, to the best of our knowledge, we have not found any previous work that focuses on the in-depth analysis of learning cases and their impact on generalization. A work in this direction was published by Berlanga et al. [6], in which their objective was to obtain robust controllers in a set of environments. The authors applied evolutionary strategies for learning navigation, observing how controller generalization decreased due to over-adaptation to training environments. Although we share the same concerns regarding generalization as Berlanga et al. [6] do, unlike them, we proposed an exhaustive and systematic analysis of the initial conditions of the training cases in the navigation environment.

Our research aims to explore how the initial conditions, including coordinates and orientation, impact the quality of controllers evolved to solve a navigation task in a particular environment. We want to investigate not only how a controller can solve the task from a specific initial condition but also how it can generalize to solve the task from 100 different, unseen initial conditions. To achieve this, we used reinforcement learning and treated each possible initial condition as a single training set, with a set of 100 unseen initial conditions serving as the test set. We refer to each training case as a "learning case" because it allows a controller to learn the rules to reach the target from that specific initial condition. We

then tested the controller on 100 different cases or samples to evaluate its generalization ability. We used an exhaustive analysis to explore all possible cases or samples in the navigation task.

The main motivation of this work is to answer the research question of wether the initial conditions (learning cases) exert by themselves a positive or negative influence on the quality of creating general controllers. Therefore, the contributions of this work are (i) proposing a methodology, (ii) performing a systematic analysis, and (iii) creating a tool to display the influence each learning case available in the navigation environment has to influence the quality of evolving general controllers. We first perform a preliminary analysis considering a reduced set of learning cases. From the results of this preliminary experiment, we hypothesized that controllers evolved from positions far from the target get more complex rules, which in turn are more robust and increase the probability of hitting the target from a broader set of initial conditions used as a test set. We then expanded our research work to perform a comprehensive analysis considering each of the positions and orientations available in the navigation environment. Experimental results from this analysis are correlated with the conventional agreement in ML that complex solutions are in general more robust.

The remainder of this paper is organized as follows: Section 2 presents some key aspects of generalization that are useful to help understand this work. Section 3 presents the basic concepts about evolutionary algorithms. Section 4 gives a brief introduction to GE. The following section, Section 5, introduces the navigation environment used in this research work, and Section 6 describes the experimental setup for both manually selected and for the exhaustive analysis, which is shown in Section 8. The main experimental results are shown and discussed as well. Finally, Section 9 presents the conclusions and future work.

## 2. Navigation Robotics

Advances in robotics have made tremendous contributions in many industrial and social domains. It is easy to note how robotics has changed the production environments and improved the industrial production. In addition, navigation robotics has been impacted by the new technologies, and currently investors focus on autonomous public and private transportation [3].

Generally, robots are created to carry out tasks that are either beyond human capability or may endanger human life. Currently, robots are not only found in industrial settings but also in homes. Their significance in the industrial environment has led to a digital transformation referred to as Industry 4.0 [7], which utilizes various technologies to enhance connectivity, efficiency, flexibility, and security in manufacturing facilities. Autonomous robots, such as autonomous mobile robots (AMRs) [8], offer benefits such as operating without human control or intervention. In the industry, AMRs are being used as cobots (coworker robots), and there is ongoing research to design robots with a smaller profile, which are less physically intimidating than large vehicles and pose less of a perceived threat. The objective is to enable human staff members to focus on tasks that only humans can perform while freeing them from tedious and repetitive activities that robots can perform better and more efficiently.

An autonomous robot is a freely moving robot that must complete certain tasks while avoiding collisions and preserving energy as much as possible without continuous human guidance, and they achieve this using the robotic hardware that is composed of the body, sensors, actuators, and the control system.

Hereinafter, we will refer to the control system as the navigation controller, or just controller for short, which provides control to an autonomous robot to process information and decide upon which actions to take. Furthermore, we limit the wide range of possible autonomous robots to a mobile robot model based on wheels, where its configuration has two independently driven wheels and one unpowered omnidirectional wheel in the rear.

Our robot has two sensors and two actuators, which are necessary to be able to perceive its environment and to move through the environment. A typical sensor is an infrared (IR) proximity sensor to compute the distance to a given obstacle. In our experiments we used three Boolean sensors that have a radius of 1-cell detection, as shown in Figure 1.

On the other hand, our robot has two servomotors as actuators, which allow it to take action, i.e., to move, turn left, or turn right. The action allowed is to move just 1-cell or turn, either left or right over the same cell where the robot is located, with the main purpose of changing its heading. Note that the controller acts over the actuators according to a set of rules that relates the information captured by the sensors to make a decision on the available choices of the actuators.

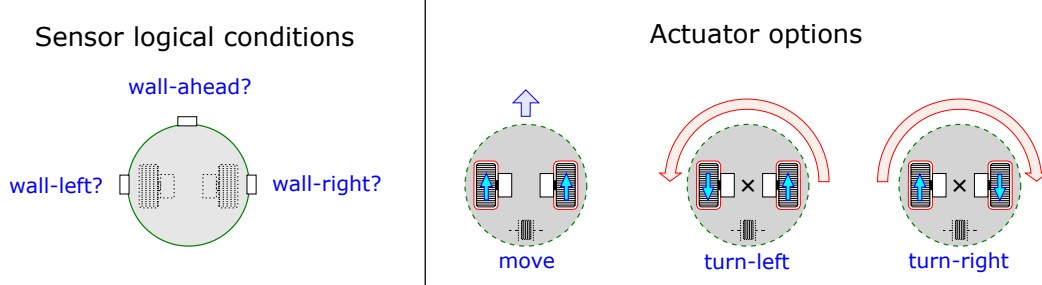

**Figure 1.** Schematic diagram of a navigation robot with two independently driven wheels and one unpowered omnidirectional wheel in the rear. At the right it shows the three Boolean sensors and at the left the three choices for the actuators.

Simulations play an important role in research and particularly in the development of autonomous robots. It is a good practice testing a robot in a simulated environment before trying the experiment in the actual environment to avoid catastrophic failure in certain situations in the real world. In addition, through simulations it is possible to test several solutions, in this case, controllers, before testing in the actual robot. Furthermore, trying experiments in the real world is very time-consuming, and generally simulations often require several iterations.

Through the years, a wide range of simulators has been proposed. In this research work, we use NetLogo [9], which is a multi-agent programmable modeling environment. The foundation of NetLogo is an agent-based modeling (ABM) paradigm, a key technique in our study, as it allows us to observe macro patterns that evolve from the interaction of micro agents (robots) and their environment (the navigation space), known as emergent phenomena, which are difficult to observe with other modeling techniques. It is also a handy environment for academic and research work well-known worldwide. Moreover, a common and successful method to generate controllers using a simulated environment is through evolutionary algorithms [10–14].

## 3. Evolutionary Robotics

The use of evolutionary principles in problem solving gave way to the field of evolutionary computing (EC) [15], which has been applied to successfully solve hard problems, not only in optimization, but also in other fields. For instance, applying EC to evolve robots created a new field named evolutionary robotics (ER), which helps to automate the process of designing robots, including morphologies and control systems [16]. Typically, ER uses Darwinian principles of natural selection to create robotic control systems.

More specifically when evolving a control system, we can typically identify four levels of organization [17], as shown in Figure 2 and described in detail in [18]: (i) genotype, (ii) phenotype, (iii) behavior, and (iv) fitness. The genotype is typically represented by a binary string and is mapped into the rules of a program. The phenotype, in our case, is seen as a program that, in turn, performs the robot's controller. The behavior is a wider concept that can be described in several ways. For our research purposes, we will understand it as a description of the robot when navigating in the environment. A shorter possible

description is using only the last or ending position of the robot in the environment after certain restrictions, for instance, its battery energy level, and this is the behavior description we use in this work. The fitness is, in our case, the robot's quality score in solving the navigation task and, following the conventional approach, it rewards the controller that gets the robot closer to the target. This, in turn, is used to guide the search through the selection of the best controllers out of a population of controllers found and understood as individuals in the artificial evolutionary process.

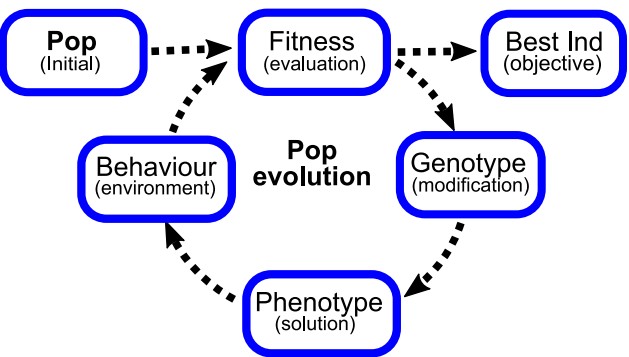

**Figure 2.** General representation for an evolutionary robotics process with four levels of organization: (1) genotype, (2) phenotype, (3) behavior, and (4) fitness.

The fitness function ideally helps to find increasingly better solutions in every iteration; typically, it is a measure of how close a particular individual, taken from the population, comes to the target.

It should be noted that prior to starting the evolutionary process, an initial step of creating the first population must be carried out, that is, the design of their genotype and phenotype. In an evolutionary algorithm, genotype and phenotype are lower and higher level descriptions, respectively, of a candidate solution (individual). The genotype is a gene string encoding a lower level abstraction of an individual, and the phenotype is composed of the features visible to a given environment.

One of the most successful methods for algorithm optimization that takes advantage of this difference and relays in a genotype–phenotype mapping process is GE, which is the method selected for our experimental setup as is explained in the following subsection.

## 4. Grammatical Evolution

Grammatical evolution [19,20], also known as GE, is a method of genetic programming, or GP [21], that utilizes a Backus–Naur form grammar, or attribute grammar [22–25], to explore the space of legal programs. It is capable of evolving computer programs or other structures that can be defined using these grammars. GE is a popular evolutionary algorithm applied to a wide range of problem domains [26–32]. The modular design of GE, shown in Figure 3, allows for the use of any search engine, typically a variable-length genetic algorithm, to evolve a population of binary strings. The individuals are then mapped onto programs using GE and evaluated using any program or algorithm.

GE programs are indirectly represented by variable length binary genomes and constructed through a developmental process. The linear representation of the genome allows for the use of genetic operators, such as crossover and mutation, in a similar way to a traditional genetic algorithm (GA), as opposed to tree-based genetic programming. The genome of each individual, encoded in codons (usually groups of 8 bits), contains the information necessary to select and apply grammar production rules, resulting in the formation of the final program, which starts with the grammar's start symbol.

The production rules for each non-terminal are assigned an index starting from 0. When selecting a production rule, beginning with the non-terminal on the left side of the program in development, the next codon value in the genome is read and interpreted using the formula $p = c \% r$, where $c$ represents the current codon value, % represents the

modulus operator, and *r* is the number of production rules for the left-most non-terminal. If the algorithm reaches the end of the genome while reading codons, it invokes a wrapping operator and continues reading from the start of the genome.

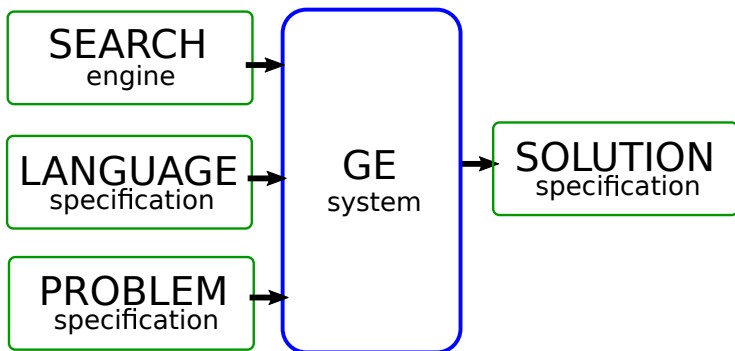

**Figure 3.** The grammatical evolution (GE) system utilizes a search engine (commonly a genetic algorithm) to generate new solutions for a specific issue by combining genetic material (genotype) and converting (mapping) it into programs (phenotype) in accordance with a language specification (interpreter/compiler).

The mapping process is illustrated with an example in Figure 4, where we use a grammar to describe maze navigation programs written in Netlogo [9].

The process ends when all non-terminal symbols have been replaced and a valid program is produced. Sometimes a *wrapping* process is used in which, if a non-terminal remains after using all the codons, the genome is reused from the beginning. If it does not replace all non-terminal symbols after a certain number of tries, it is considered invalid and given the lowest possible fitness. An example of this process is shown in the Figure 4, which uses a grammar to create programs for navigating mazes in Netlogo [9].

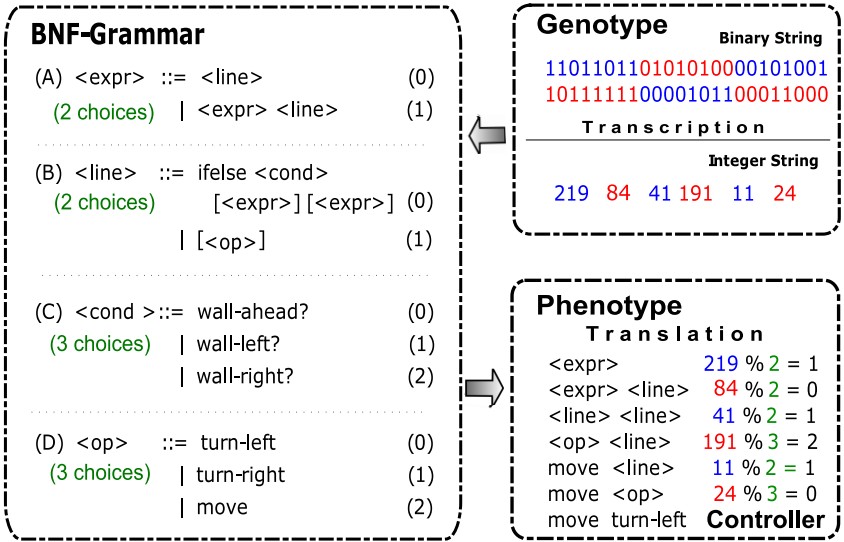

**Figure 4.** A process of mapping a genotype (binary) to a phenotype (controller/program) using genetic encoding (GE) is demonstrated. The binary genotype is divided into segments called codons, which are then transcribed into an integer string. This integer string is then used to choose production rules from a predefined grammar, which are then translated into a sequence of actions (controller).

## 5. Case Study

In the field of ML for autonomous agent navigation, most studies do not use fixed training sets when using evolutionary computation. Instead, general controllers are typically developed by training on a dynamic set of instances.

This research aims to investigate initial conditions that a robot can use in a given navigation task to evolve general controllers through reinforcement learning. Each initial condition is treated as a single training case, allowing us to identify which positions in the navigation space are better suited for evolving better general controllers. There is evidence in [33] that using a single instance helps to balance learning and overfitting. This study chose each available position and orientation as a training case and a set of 100 cases for testing.

In this research, we use a rectangular grid of $38 \times 22$ cells as the navigation environment, with $37 \times 22$ cells if the outer walls are not taken into account. The target is represented by a red square that is enclosed by a U-shaped wall, as depicted in Figure 5. This navigation environment is comparable to the one studied previously in [34] to try a divergent set, which was referred to as the medium maze, and [35] where authors presented an extensive study of generalization in a GE-based evolutionary learning system.

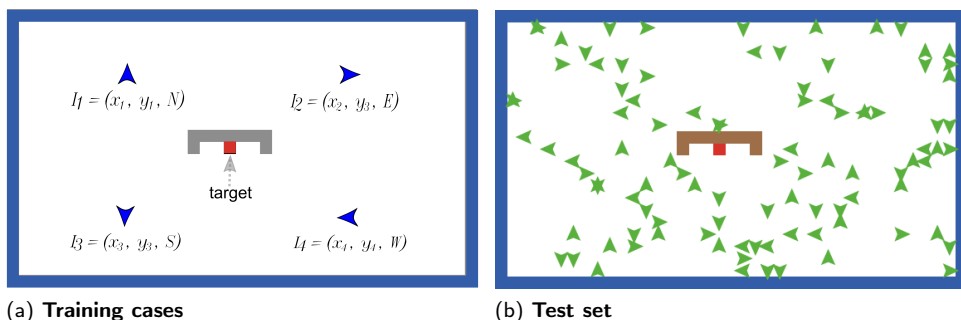

(a) **Training cases**　　　　　　　　　(b) **Test set**

**Figure 5.** (**a**) A simple navigation task showing a small set of possible cases that could be used as a training set to reach the target (the red square at the center). (**b**) A set of 100 cases as the test set fixed for all the experiments.

Each training example in this scenario is described by a triple of information, $I_i = (x_i, y_i, \theta_i)$, which specifies the starting position of the agent within a grid environment, given by the row $x$, column $y$, and the initial orientation $\theta_i$, which can be one of four options: north (N), south (S), west (W), or east (E). Figure 5a illustrates a set of four possible cases, $I_1 - I_4$, around the target.

If the problem requires specific initial conditions then this is not an issue, but determining how to construct the training and test set for an arbitrarily complex environment is in no way a trivial task. In this work, we constructed the test set in a similar way as in [35], following a predefined research line to later use other strategies to guide the search, such as novelty search [34], and being consistent with this previous work to make further comparisons. Therefore, we sampled the entire navigation environment to randomly select 100 initial conditions, each with a position and an orientation to come up with the configuration shown in Figure 5b. This test set was held constant for all experiments, including the preliminary and exhaustive analysis. Additionally, we took care that the training and test sets have no overlap.

## 6. Experimental Setup

This section explains the experimental work that we have proposed to examine the effect of learning cases on enhancing the generalization capability of a system that is based on GE.

A high level view of our experimental work is depicted in Figure 6, consisting of: (1) a navigation task; (2) choosing an initial orientation for each case; (3) performing an exhaustive selection for each cell as a learning case in the navigation environment; (4) running an evolutionary algorithm 30 times to evolve navigation controllers, in this case GE; (5) choosing the best controller from each run and testing it over the test set; and, finally, (6) performing an overfitting measure by computing the difference between the training and test scores.

The parameters used in the experiments are summarized in Table 1 and, as our focus is not to test a set of different parameters, we set them to be as general as possible. We are following a similar parameter setup from [35] for a further comparison. Furthermore, we are using the original codon-size parameter from [19] with a minimum size of 15 and a maximum size of 25, and the number of wraps allowed is 10. Because we are using the GE implementation from [36], we keep its setup for the genetic operations, where for both crossover and mutation we use the one point type, and the rates are 0.90 and 0.01, correspondingly. For the evolution, we use a generational strategy, and for the parent selection, we use the roulette. The number of runs is set to 100 instead of a more conventional number of 30 to reduce as much as possible the uncertainty on the experimental results, and, finally, the population is set to 250 individuals each run.

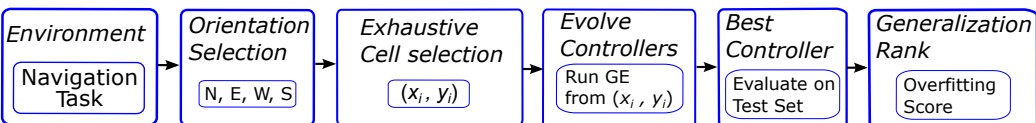

**Figure 6.** High level description of the proposed experimental work for both manually selected and exhaustive experiments.

**Table 1.** Parameters used for the experimental work. Codons-min and Codons-max are the minimal and maximal number of codons in the initial random population of the GE search.

| Parameter | Value | Parameter | Value |
| --- | --- | --- | --- |
| Codon-size | 8 | Codons-min | 15 |
| Codons-max | 25 | Wraps | 10 |
| Crossover | codon | X-over prob. | 0.9 |
| Crossover type | One Point | Mutation type | One Point |
| Mutation prob | 0.01 | Elitism | 10% |
| Generational | YES | Selection | Roulette |
| Runs | 100 | Individuals | 250 |

The function used to evaluate the quality of each controller assigns a quality score to each agent $\psi_i$ by computing the inverse of the Euclidean distance taken from the average of the agent's final positions to the target $t$, as follows:

$$F(\psi_i) = \frac{1}{1 + \text{dist}(\beta_i, t)} \tag{1}$$

where $\beta_i$ is the average from the agent's final positions $\alpha_1^i, \ldots, \alpha_m^i$, reached from each of the learning instances used as training set.

This function used in the learning process with the training set is known as the fitness function to assign a fitness score to each controller. The fitness function is used to drive the algorithm in the search space looking for the best controllers. In a 2D space, the standard metric to use is the Euclidean distance [34,35]; this metric measures how far a robot is from the target, so, using the inverse of the Euclidean distance, the fitness function rewards controllers that allow robots get closer to the target. A usual method is to consider the final position from the robot taking into account the maximum movements allowed, which can be understood as the maximum energy allowed when using a battery.

## 7. Preliminary Analysis

The topic of generalization in a GE-based evolutionary learning system was thoroughly examined in [35]. The main focus of the research was to investigate the impact that the training set has on the ability of a evolutionary algorithm system to evolve effective controllers. The study specifically looked at two factors related to the training set, including its size and the method used to select instances from the overall set of possibilities. The

size of the training set was varied, starting from a single instance and reaching up to 60 instances. Two different methods for selecting instances were also examined, including random selection and manual selection.

In this initial experiment, we assess the bias that a human designer introduces into the learning process. To accomplish this, a group of 12 examples shown in Table 2, were intentionally picked, comprising positions from three different corners, as well as directly under the target: two close to it and one behind the wall close to the target.

**Table 2.** List of 12 learning cases manually selected for the preliminary analysis, coordinates are refered to the origin located at the left bottom of the navigation environment and are depicted in the Figure 7.

| Learning Cases | Coordinates | | Orientation |
| :---: | :---: | :---: | :---: |
| $I_p$ | Horizontal | Vertical | |
| $I_1$ | 5 | 6 | North |
| $I_2$ | 5 | 6 | East |
| $I_3$ | 1 | 22 | East |
| $I_4$ | 1 | 22 | South |
| $I_5$ | 17 | 13 | North |
| $I_6$ | 17 | 13 | South |
| $I_7$ | 16 | 11 | West |
| $I_8$ | 16 | 11 | East |
| $I_9$ | 19 | 1 | North |
| $I_{10}$ | 19 | 1 | West |
| $I_{11}$ | 31 | 18 | West |
| $I_{12}$ | 31 | 18 | North |

The overall locations of this group are illustrated in Figure 7, where the table sorts all learning cases used for training and the figure shows their location graphically. It must be noted that we have six different locations with two different orientations each.

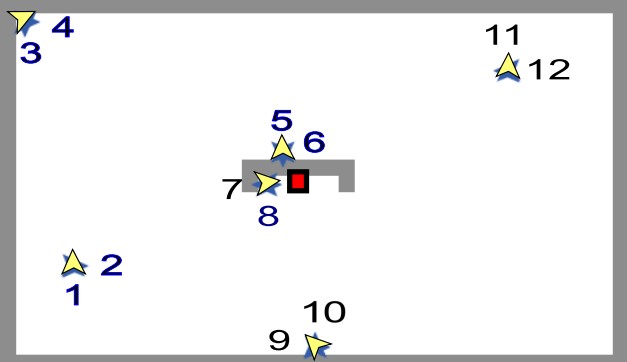

**Figure 7.** Learning cases for the preliminary analysis, manually selected as initial conditions to locate each robot in the navigation environment, graphically depicted by yellow arrows showing their location and orientation and in red at the middle is the target.

For this experimental setup, each of the 12 examples, manually selected, was treated as a separate training set, i.e., each training set contained only that example. This provided a clear understanding of how that specific training example affected learning. As seen in the experimental results in [35], it is clear that using just a single example in this manner does not easily result in a general solution, which was expected, but the study did suggest that the location in the navigation environment plays a crucial role in guiding the search to find general solutions, and this is the core idea to perform a comprehensive individual analysis.

Table 3 presents a summary of the average performance of the optimal solution using each of the 12 manually selected cases $I_p$.

**Table 3.** Experimental results from the preliminary analysis using 12 learning cases manually selected, which are sorted according to their 'Test Score' shown in bold and they are grouped into two sets: in blue are the cases that get higher scores and, in red, the cases with lower scores.

| Learning Cases | | Training | Test | | Overfiting |
| Rank | $I_p$ | Score | Score | Hit | Train-Test |
|---|---|---|---|---|---|
| 1 | $I_{11}$ | 1.00 | **0.665** | 3% | 0.335 |
| 2 | $I_2$ | 0.67 | **0.422** | 2% | 0.248 |
| 3 | $I_1$ | 0.64 | **0.387** | 6% | 0.253 |
| 4 | $I_3$ | 0.69 | **0.328** | 1% | 0.362 |
| 5 | $I_{12}$ | 0.74 | **0.286** | 3% | 0.454 |
| 6 | $I_4$ | 0.95 | **0.201** | 2% | 0.749 |
| 7 | $I_5$ | 0.99 | **0.125** | 0% | 0.865 |
| 8 | $I_6$ | 1.00 | **0.120** | 0% | 0.880 |
| 9 | $I_9$ | 0.95 | **0.107** | 2% | 0.843 |
| 10 | $I_{10}$ | 1.00 | **0.096** | 0% | 0.904 |
| 11 | $I_7$ | 1.00 | **0.081** | 0% | 0.919 |
| 12 | $I_8$ | 0.98 | **0.072** | 0% | 0.908 |

To compare the impact of the bias introduced by each learning case on the ability of the evolved controllers to generalize to 100 previously unseen cases, they are ranked (Rank) based on their test score, which is highlighted in bold in the fourth column as Test Score. The number of times the target was successfully reached is also included as Hit, and Overfitting is a measure of how much the individuals have been over-trained.

When focusing only on the scores from training, we noticed that most of them were high scores, but when we sorted the instances according to the test score, then we noticed some interesting trends, for instance, the positions close to the target allowed to evolve controllers with a more simple structure in the conditional rules, which struggle to reach the target from the majority of the set of 100 test cases. Almost all of them could not solve the task from any of the test cases, whereas the positions far from the target were a better choice to evolve general controllers.

As a naïve example to explain this behavior, if we choose one of the positions right below the target and also with its orientation towards the target, the population in the evolutionary process will converge rapidly to a set of simple rules as a controller to reach the target from that position. One possible controller could be so simple to contain the single choice of moving forward (move). This controller will have an excellent score in training but a poor performance in test, because it does not consider avoiding any walls. We could say, in this case, that this controller overfit the training set. From this preliminary experiment, we could hypothesize that controllers evolved from positions far from the target get more complex rules, which in turn are more robust and increase the probability of reaching the target from a wider set of of initial conditions.

One interesting observation we can note from Table 3 is that the first half of cases are located far from the target (in blue), whereas the second half are close to the target (in red), as illustrated in Figure 8, particularly the learning cases $I_5$ to $I_8$ surrounding it. This observation gives us a clue about the existence of regions that positively influence the search algorithm, in this case GE, to evolve general capabilities in the controllers and similarly to assume there must exist other regions where their impact is negative.

Manual selection undoubtedly introduces a human bias in the analysis that could compromise the conclusions drawn from the experimental results. The next step is to perform a comprehensive analysis on the entire navigation task to the verify the trend found in the preliminary experiment.

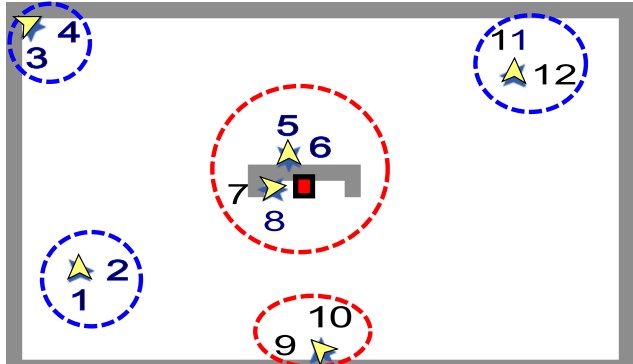

**Figure 8.** Learning cases are grouped into two sets according the evaluation to generalise using the test set, in blue are the cases with higher scores and, in red the cases with lower scores.

## 8. Exhaustive Analysis

Motivated by the insights from the preliminary experiment using manually selected learning cases, we performed an exhaustive experimentation taking each cell in the navigation environment as a possible learning case, excluding walls and the target. In this experiment we are still using the same experimental setup, as shown in the Figure 6; the same set of test cases, as in the preliminary experiment; and the parameters for GE. Figure 9 illustrates four sets of learning cases, according to the available orientations (north, east, south, and west). Every cell in the navigation environment is taken as the coordinates to exhaustively select them as a learning case each, and we run a GE-system 30 times to get an average out of the scores from the best controllers found in each run.

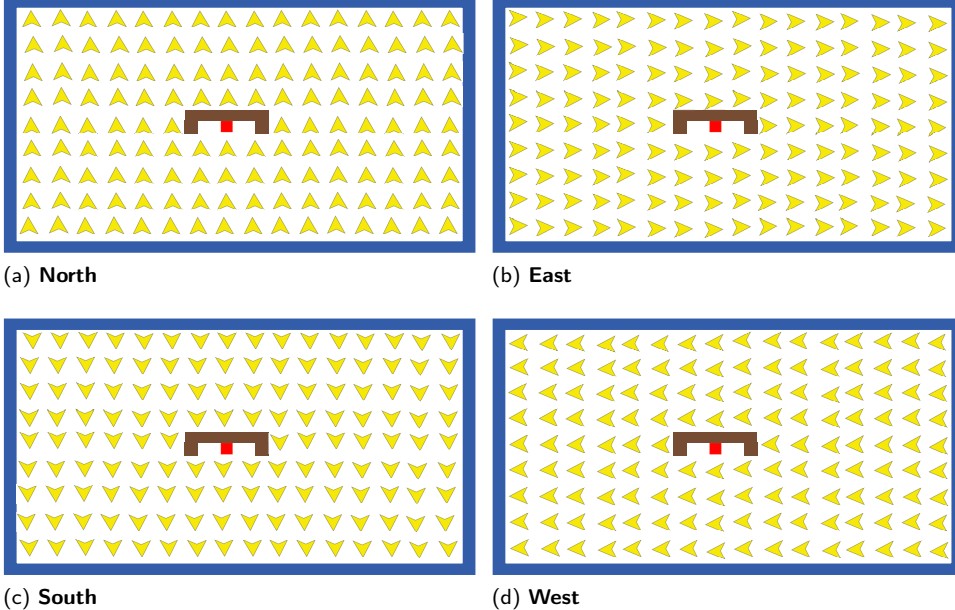

(a) **North**

(b) **East**

(c) **South**

(d) **West**

**Figure 9.** Representation of all locations in the navigation environment where a GE-system was run 30 times from each of four different orientations.

To the best of our knowledge, previous studies have not performed an exhaustive analysis of the effects that each initial condition, training instance, or learning case has on generalization for a navigation problem in ER. Because we are interested in evolving general controllers, the measure employed to evaluate the generalization capabilities of the GE-system is the overfitting, understood as the difference between the test and training scores computed by using the function $F(\psi)$ shown in Equation (1), and the overfitting output value is in the range of $[0, 1]$. Using all the overfitting values, we constructed four heat-maps from each different orientation (north, south, west, and east), as illustrated in

Figure 10, to graphically observe the results from this exhaustive experimentation. We can clearly see how two different regions emerge from these heat-maps. Regions tending to red negatively influence developing general capabilities in the controllers evolved by the GE-system, whereas regions tending to blue contain learning cases that highly increase the possibilities to evolve general controllers from their coordinates. The orientations do not seem to play an important role, although there are certain graphical differences, particularly at the bottom below the target.

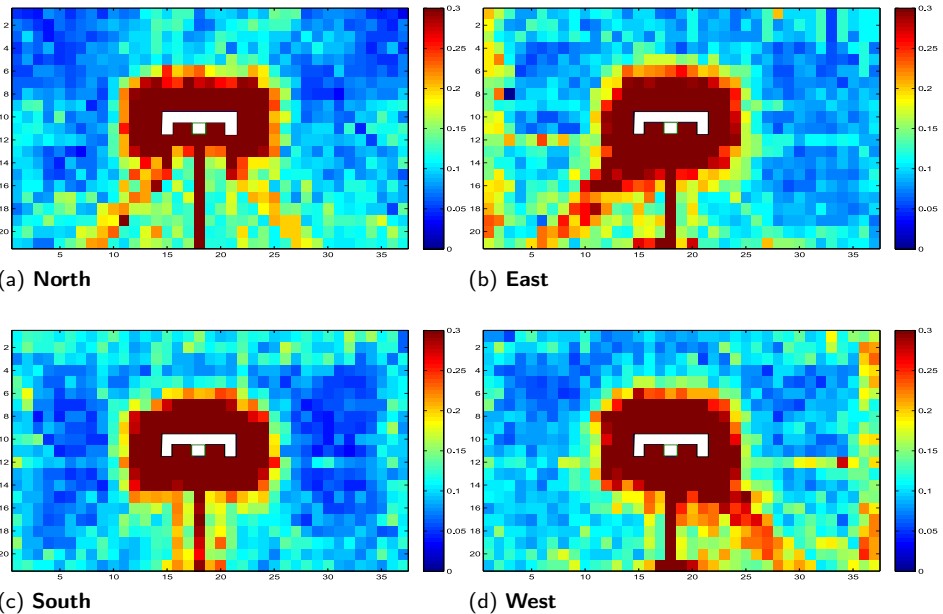

(a) **North**

(b) **East**

(c) **South**

(d) **West**

**Figure 10.** Heat-maps show the overfitting performance of each potential training instance's starting point. Blue cells indicate areas where GE tends to produce better general controllers, whereas red cells indicate areas where GE tends to produce less effective general controllers.

It can be noted that the observation from the preliminary experimentation is replicated on this exhaustive experimentation, and we can conclude that, for the navigation environment selected, the GE-system prefers locations far from the target to evolve general controllers. We cannot assume that these observations and conclusions can be overlapped to different navigation tasks, or even to the performance from other search algorithms; these are research questions that must be addressed in future work. One possible explanation about the existence of these regions is that locations close to the target used as training cases evolve simpler controllers because the agent (robot) quickly find the target in the learning process; when these controllers are tested from the set of 100 unseen cases, they really struggle to reach the target. On the other hand, we suppose that GE struggle in the training process to evolve controllers that can reach the target from locations far away from the target, meaning that must find more complex controllers, which in turn are more robust and intrinsically contain more general behaviors to reach the target from most of the test cases used. This observation gives us another possible venue for future work, where now we could focus on analysing the controllers to find any pattern that perhaps relates their behaviors with their generalization capabilities.

## 9. Conclusions

The aim of this research is to explore how the characteristics of a navigation task itself affects the evolution of general controllers. In this work, we focused on the position and orientation as initial conditions for a robot to navigate in the given environment. We proposed a methodology to perform a comprehensive analysis of the initial conditions as learning cases to solve a navigation task, where each case is seen as a combination of an initial position and orientation in the navigation environment.

We first performed a preliminary experiment with a set of small learning cases manually selected. We choose six initial locations with two different orientations each, having twelve learning cases in total, using each of them as a single training case and a hundred unseen initial conditions as the test set. From this preliminary experiment, the main observation, after sorting out the cases according to the test score, is that the half of the cases located far from the target showed better performance in test, and the second half close to the target show not as good performance as the first half. This observation suggests to us the existence of regions that positively influence the search algorithms to evolve general capabilities in the controllers.

We then extended our research to perform an exhaustive analysis, considering now each position and orientation available in the navigation task. The second experiment consisted of a comprehensive analysis in the same navigation task, taking each of the available cells as the initial locations using four orientations, each combination of location and orientation was used as a learning case, where a GE was executed 100 times to evolve controllers from these initial conditions to compute the training and test scores. From their difference, we obtained the overfitting score. For each learning case, the average of the 100 overfitting scores was used to construct a heat-map to visualize the influence of each learning case on GE to evolve general controllers.

From this experiment and analysis, we can note how the learning cases are grouped into two groups in a similar way as in the preliminary experiment using just 12 learning cases, where the cases far away from the target positively influence the development of generalization capabilities in the controllers. This observation is correlated with the conventional agreement in ML that complex solutions are, in general, more robust. Therefore, from this comprehensive analysis, we learned that for this particular navigation task, the locations close to the target are not good to use as learning cases to evolve general solutions. The experimental results do not give us the information to answer the question related to how far the locations must be to be considered good cases.

Even though performing an exhaustive analysis is expensive, it is worth knowing how the locations more than the orientations impact the evolution of general controllers. For this reason, we plan future work to (i) extend our research line to verify other navigation tasks if there are regions that positively impact the evolution of general controllers, (ii) study how a combination of initial conditions taken from different regions (close and far) impact the evolution of general controllers, (iii) focus on the controller structure (set of rules) to differentiate the simple from the complex and look for relationships with the positions in the navigation environment.

**Author Contributions:** Conceptualization, E.N. and P.U.; methodology, E.N., C.S. and P.U.; validation, E.N., C.S., P.U., F.G. and F.L.; formal analysis, E.N. and F.G.; investigation, E.N., C.S., F.G. and F.L.; writing—original draft preparation, E.N., C.S., F.G. and F.L.; writing—review and editing, E.N., C.S., F.G., F.L., P.U., L.T. and C.R. All authors have read and agreed to the published version of the manuscript.

**Funding:** Enrique Naredo acknowledge Lero—the Irish Software Research Centre (www.lero.ie), Leonardo Trujillo was supported by the project 14271.22-P funded by Tecnológico Nacional de México (TecNM), and Conor Ryan is partly supported by the Science Foundation of Ireland Award 16/IA/4605.

**Conflicts of Interest:** The authors declare no conflict of interest.

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
