# Peer review of "Comprehensive Analysis of Learning Cases in an Autonomous Navigation Task for the Evolution of General Controllers"

_mca, doi:10.3390/mca28020035_

Round 1

Reviewer 1 Report

The paper entitled 'Comprehensive Analysis of Learning Cases in an Autonomous Navigation Task for the Evolution of General Controllers' investigate the impact of training scenarios on the learning process for autonomous navigation tasks. In my opinion, the structure of the work is appropriate and the literature review is sufficient. The topic of the work is also interesting, but I'm not sure if the work is innovative enough. Additionally, I have some reservations:

1. There are a few minor language errors in the work (for example, despite the introduction of the acronym ML, the name 'machine learning' is used many times).

2. Many papers are not cited in the text (e.g. [32]).

3. There is no information on how the values of the parameters presented in Table 1 were determined. The mere statement 'which are standard values for GE systems' is not sufficient - at least references to the literature are missing.

4. What mutation and crossover operators have been used?

Author Response

Dear Reviewer-1

the authors really appreciate the time you spend in giving comments and feedback about our manuscript, below you will find the answers to all the questions (in blue) you provided.

1. There are a few minor language errors in the work (for example, despite the introduction of the acronym ML, the name 'machine learning' is used many times).

We thank the reviewer for this comment and we already verified that all occurrences of the name 'machine learning' after being introduced its acronym are replaced by 'ML.' 

2. Many papers are not cited in the text (e.g. [32]).

Thanks to the reviewer for pointing out this error that occurred from not being careful to remove the references while updating the manuscript. We have already eliminated all the references not cited in the manuscript, we have classified them all to have an ordered citation as they are cited and we will add other references to comply with some comments and suggestions from other reviewers.

3. There is no information on how the values of the parameters presented in Table 1 were determined. The mere statement 'which are standard values for GE systems' is not sufficient - at least references to the literature are missing.

Authors will add in the manuscript a discussion related to the parameters used in the experiments for GE. On one hand, will add a reference of a previous work where the authors performed a preliminary parameters selection and we are following this setup as a reference and further comparison. Furthermore, authors will add some references to support the parameter selection used in other domains. In general, most of the GE implementations as the Java  implementation for Netlogo (https://www.ijcai.org/Proceedings/11/Papers/214.pdf), which we are using in this work, follow the original idea of using 8 bits for the codon size, the range (min & max) of numbers of codons refer to the complexity of the solution (individual) proposed by GE, usually is allowed to GE to take wrap) same codons up to 10 times, genetic operations are similar to other evolutionary algorithms, having 90% for crossover and 1% for mutation, crossover usually  takes a value from a range of 70% to 90%, and mutation takes low rates, but could increase to have say 5%, meaning that we increase of exploration, some works deal with the exploration-exploitation issue by using higher rates at the beginning and low rates at the end, in our work we are not interested by now in researching how these parameter impact in the navigation task (which is an interesting research question), but rather we tried to keep a more general, such as the generational and roulette choices and in this sense 'standard' setup to get less uncertainty from other variable in our research work. On the other hand, since we are performing an exhaustive analysis, so we decided to do 100 instead of 30 runs.

4. What mutation and crossover operators have been used

The implementation of JGE provides a One Point Crossover as well as a One Point Mutation, which are the choices the authors used in this work. As stated before we have chosen these options to keep our set-up as general as possible to focus only on the generalisation task and how each initial condition impacts on the quality of the evolved controllers. We will add this information in the manuscript.

Hope we answered all the questions and solved any doubt about the importance and impact of our research work in our scientific community.

Sincerely,

The Authors

Reviewer 2 Report

The focus of this research is to investigate the impact of training scenarios on the learning process for autonomous navigation tasks. Specifically, they conducted a comprehensive analysis of a particular navigation environment, using evolutionary computing to develop controllers for a robot, starting from different locations and aiming to reach a specific target. The final controller was then tested on a large number of unseen test cases. Although the validity of this method has been proved by experiments, there are still some problems. The following suggestions could be of help to further improve the quality of this paper.

Q1: Introduction: What are the advantages of robots? Why do robots appear in our lives? What are the benefits of navigating robots? Please elaborate.

Q2:Introduction: The paper gives four innovation points, and the first three innovation points are all about case analysis. Does it mean that there are different analysis methods for different cases? If yes, please elaborate. If not, please redescribe these three innovations.

Q3:Navigation Robotics: This part does not explain much about related applications of Navigation Robotics, and there are also few references. Please add.

Q4: 3. Evolutionary Robotics: Only one subchapter of the chapter is inappropriate. This part can be divided into two parts, or can be directly discussed as a whole. Please modify.

Q5: 4. Case Study: The article mentions the use of basic navigation tasks and the possibility of selecting only one training set. How does this benefit the study of this article? Please elaborate.

Q6: What is the meaning of Formula 1? How is the evaluation function evaluated? Please elaborate.

Q7: Preliminary Analysis : Please explain why the results in some test cases are poor.

Q8: Conclusions: What is the motivation for the approach presented in the article? What are the disadvantages of the method? Please add.

Author Response

Dear Reviewer-2

the authors really appreciate the time you spend in giving comments and feedback about our manuscript, below you will find the answers to all the questions  (in blue) you provided.

Q1: Introduction: What are the advantages of robots? Why do robots appear in our lives? What are the benefits of navigating robots? Please elaborate.

Robots, in general, are designed to do many things humans are not able to do or as a resource to prevent a human life under a risk. Nowadays, robots are not only present at the industrial environment but also in our homes. Robots are so important in the industrial environment today that we have together with other technologies a digital transformation named as Industry 4.0, which help keep the manufacturing facilities connected, efficient, flexible & protected. 

The benefits of autonomous robots are, for instance, for an autonomous mobile robots (AMRs) that they operate without human control or intervention. The trend about AMRs in the industry is to use them as cobots (coworker robots), there is an active research in this field to design robots with a small profile to be less physically intimidating than large vehicles and less of a perceived danger. One of the goals about using AMRs is to help ensure the human staff members do the things only humans can do and liberating them from many of the tedious and repetitive activities that a robot can perform better and quickly.

We will add some references about these facts in the manuscript to highlight the robots' importance.

Q2: Introduction: The paper gives four innovation points, and the first three innovation points are all about case analysis. Does it mean that there are different analysis methods for different cases? If yes, please elaborate. If not, please redescribe these three innovations.

The main contributions of our work is related to analysing the impact on the quality of generalising by the controllers evolved from each possible initial condition (coordinates & orientation) in a given navigation environment. The research question we addressed is: how each initial condition impacts on the quality, not only in solving the navigation task by evolving a controller, but rather to evolve a controller which can reach the target from other 100 unseen initial conditions addressing this task as a generalisation problem? In order to perform a reinforcement learning for the generalisation task, we considered each of the possible initial conditions (cell & orientation) as a single training set and a set of 100 unseen initial conditions as the test set. Each single case is seen as a 'training case' and since this is used in the learning process, similar as in any machine learning process, we can name it as a 'learning case', which is a sample of the entire environment, in this sense we could also use 'sample analysis', but because we are making a difference between the training and test set used, we use 'learning case' as a single training case to allow a controller learn the rules to reach the target from that particular case or sample, and then we test that controller from 100 different cases or samples. Following this idea, we do not propose a new method to perform an analysis, but rather using a well-known method, such as an exhaustive analysis, we analyse all possible cases or samples from the given navigation task. We will add some references to clarify these ideas to the reader, using some examples from other domains, such as classification and about the importance of doing an exhaustive analysis. 

Q3: Navigation Robotics: This part does not explain much about related applications of Navigation Robotics, and there are also few references. Please add.

In line with the question 1 (Q1), we will add more references in the manuscript about related applications of Navigation Robotics from recent years and even from some of the past to highlight how this still is a very important research field related to Artificial Intelligence in collaboration with other fields, such as machine learning, computer vision, deep learning, and impacting in the real-world activities such as the functions described in the Industry 4.0.

Q4: 3. Evolutionary Robotics: Only one subchapter of the chapter is inappropriate. This part can be divided into two parts, or can be directly discussed as a whole. Please modify.

Thanks for the observation and suggestion, the authors have decided to convert the 'Grammatical Evolution' subsection into a section, this way we will have sections "3. Evolutionary Robotics" and "4. Grammatical Evolution."

Q5: 4. Case Study: The article mentions the use of basic navigation tasks and the possibility of selecting only one training set. How does this benefit the study of this article? Please elaborate.

The main focus of this research work is to study each single case that a robot can take in a given navigation task to evolve general controllers, in the reinforcement learning, each case can be seen as a single training case, the main benefit of using this approach is to identify how certain positions grouped into regions in the navigation space, where they could be better to choose as training cases to evolve better general controllers. We will add a reference related from other domain entitled "Balancing Learning and Overfitting in Genetic Programming with Interleaved Sampling of Training Data" where the authors proposed to use a single training instance (case) at each generation and balance it with periodically using all training data. The motivation for this approach is based on trying to keep overfitting low (represented by using a single training instance), even though our research work does not consider an interleaved method we considered each cell (and orientation) to perform an exhaustive analysis, in a future work we could consider a subset of instances as a training set.

Q6: What is the meaning of Formula 1? How is the evaluation function evaluated? Please elaborate.

The Equation 1, describes a function used to evaluate each controller and to assign a score related to the quality of solving the navigation task, this function used in the learning process with the training set is known as the 'fitness function', which in turns assigns the fitness score to each controller. The fitness function is used to drive the algorithm in the search space looking for the best controllers. In a 2D space the standard metric to use is the Euclidean distance, this metric measures how far a robot is from the target, so using the inverse of the Euclidean distance, the fitness function rewards controllers which allow robots get closer to the target. An usual method is to consider the final position from the robot taking into account the maximum movements allowed, which can be understood as the maximum energy allowed when using a battery. We will add some references in the manuscript to allow all readers understand this evaluation process.

Q7: Preliminary Analysis : Please explain why the results in some test cases are poor.

We initially performed an experiment with a small set of manually selected initial conditions used each as a single training instance to get a flavour of the feasibility of doing an exhaustive analysis. In Figure 8, we can see a table and a subfigure with the results from this experiment. At the beginning, when focusing only on the scores from training, we noticed that most of them got high scores,  but when we sorted the instances according to the test score, then we noticed some interesting trends, for instance, the positions close to the target allowed to evolve controllers with a more simple structure in the conditional rules, which struggle to reach the target from the majority of the set of 100 test cases, almost all of them couldn't solve the task from any of those test cases, whereas the positions far from the target (considering the specific given navigation task), were a better choice to evolve general controllers. As a naive example, we could choose one of the positions right below the target and with the orientation towards the target, this instance will rapidly find almost any set of rules as a controller to reach the target from that position, in such a way that one possible controller could be so simple that could contain the single choice of moving forward (move), but when testing this simple controller from all 100 test cases, surely will have one of the worst scores because does not consider avoiding any wall. Following this idea, we can now understand that controllers evolved from positions far from the target get more complex rules, which in turn are more robust and increase the probability of reaching the target from a wider set of of initial conditions. We will add this discussion in the manuscript to allow any reader a better understanding of the preliminary experimental results.

Q8: Conclusions: What is the motivation for the approach presented in the article? What are the disadvantages of the method? Please add.

The motivation is to research about how a navigation task impacts by itself to evolve general controllers and to start a research line towards looking for patterns in the navigation tasks which can perform some kind of regions and researchers can then focus first in looking for these regions and then test any algorithm. The disadvantages of this research work, are 1) it is computational expensive (as any exhaustive analysis), if we try now a more complicated navigation task the time will surely grow exponentially, 2) it is not easy to compare the results because we couldn't find any similar work in the literature (which in turn could be a benefit), 3) there are other variable and parameters that we could try to control to find any relationship with the influence from any initial condition (this is part of our research line, but not part of this research work). We will add a discussion about the advantages and disadvantages of our work.

Hope we answered all the questions and solved any doubt about the importance and impact of our research work in our scientific community.

Sincerely,

The Authors

Reviewer 3 Report

This study investigates the impact of training scenarios on the learning process for autonomous navigation tasks. The authors carry out an exhaustive analysis of a particular navigation environment, using evolutionary computing to develop controllers for a robot, starting from different locations and aiming to reach a specific target. The final controller was then tested on a large number of unseen test cases and experimental results provide strong evidence that the initial selection of the learning cases plays a role in evolving general controllers.

The paper is well constructed. It has an adequate introduction, coupled with a presentation of the state-of-the-art in the field. The methodology section is extensive, and the results show the capabilities of the method proposed by the authors. I found this paper so interesting, it provides a useful contribution to its area of research.

 However, a several details need to be corrected:

- What are the limitation(s) of the methodology adopted in this work?

-In lines 332-339 of the conclusions I understand the authors reflect future work of this research and should be better redacted

-The sequence used to reference the bibliography is not common. It is usual to start with [1]...

-Reference 60 is incomplete

-Please check and correct errors in the bibliography

Author Response

Dear Reviewer-3

the authors really appreciate the time you spend in giving comments and feedback about our manuscript, below you will find the answers to all the questions  (in blue) you provided.

1. What are the limitation(s) of the methodology adopted in this work?

The limitations of this research work, are 1) it is computational expensive (as any exhaustive analysis), if we try now a more complicated navigation task the time will surely grow exponentially, 2) it is not easy to compare the results because we couldn't find any similar work in the literature (which in turn could be a benefit), 3) there are other variable and parameters that we could try to control to find any relationship with the influence from any initial condition (this is part of our research line, but not part of this research work). On the other hand, the advantages are 1) to study each single case that a robot can take in a given navigation task to evolve general controllers, 2) to identify positions in the navigation space, which are better to select as training instances. We will add a discussion about the advantages and disadvantages of our work.

2. In lines 332-339 of the conclusions I understand the authors reflect future work of this research and should be better redacted.

Many thanks for pointing out this error, we will rewrite this part to clearly define our future work. We will focus  on 1) extending this actual work to verify first if there is any region in the actual navigation task which impacts positively to evolve general controllers, 2) to study how a combination of initial conditions taken from different regions are help to evolve general controllers, 3) using other more complex navigation task to analyse if we can observe similar trends, 4) to focus on the controller structure (set of rules) to look for any relationship with a set of initial conditions. 

3. The sequence used to reference the bibliography is not common. It is usual to start with [1]...

Thanks for the observation, we already solved this issue and got a sorted list of references. 

4. Reference 60 is incomplete

Many thanks for this observation, we already performed an exhaustive revision to get all of our references complete.

5. Please check and correct errors in the bibliography

Thanks again for this observation, authors already got a new version with the correct bibliography, sorted and with complete references.

Hope we answered all the questions and solved any doubt about the importance and impact of our research work in our scientific community.

Sincerely,

The Authors

Round 2

Reviewer 1 Report

The authors took into account my comments, therefore I recommend accepting the paper.

Author Response

Dear Reviewer-1

the authors thank you for accepting our manuscript to be published in this special issue of the Mathematical and Computational Applications (MCA) journal part of the Multidisciplinary Digital Publishing Institute (MDPI).

Sincerely,

The Authors

Reviewer 2 Report

Q1: Abstract: the motivation of the method proposed is lack. Please elaborate.

Q2:Introduction: Lacking the description of motivation in introducing the innovation. Please add.

Q3: How to get the testing scenarios used in experiment? Why did you choose these scenarios? Please elaborate.

Author Response

Dear Reviewer-2

the authors thank you for the further comments and observations, which help us to improve the writing of our manuscript. Here you have the answers to each of the questions.

Q1: Abstract: the motivation of the method proposed is lack. Please elaborate.

To introduce correctly the motivation, we rewrote part of the Abstract:

"This research aims to explore how training scenarios affect the learning process for autonomous navigation tasks.  The primary objective is to address whether the initial conditions (learning cases) have a positive or negative impact on the ability to develop general controllers.  By examining this research question, the study seeks to provide insights into how to optimize the training process for autonomous navigation tasks, ultimately improving the quality of the controllers that are developed.  Through this investigation, the study aims to contribute to the broader goal of advancing the field of autonomous navigation and developing more sophisticated and effective autonomous systems."

Q2:Introduction: Lacking the description of motivation in introducing the innovation. Please add.

We rewrote a paragraph  in the Introduction section to describe the motivation before introducing the contributions: 

"The main motivation of this work is to answer the research question of wether the initial conditions (learning cases) exert by themselves a positive or negative influence on the quality of creating general controllers.
Therefore, the contributions of this work are (i) proposing a methodology, (ii) performing a systematic analysis, and (iii) creating a tool to display the influence each learning case available in the navigation environment has to influence the quality of evolving general controllers.
We first perform a preliminary analysis considering a reduced set of learning cases. From the results of this preliminary experiment, we hypothesized that controllers evolved from positions far from the target get more complex rules, which in turn are more robust and increase the probability of hitting the target from a broader set of initial conditions used as a test set.
We then expanded our research work to perform a comprehensive analysis considering each of the positions and orientations available in the navigation environment. Experimental results from this analysis are correlated with the conventional agreement in ML that complex solutions are in general more robust."

Q3: How to get the testing scenarios used in experiment? Why did you choose these scenarios? Please elaborate.

We already added a paragraph discussing how to get the testing scenarios and why we have chosen them:

"If the problem requires specific initial conditions then this is not an issue, but determining how to construct the training and test set for an arbitrarily complex environment is in no way a trivial task. In this work, we constructed the test set in a similar way as in [35], following a predefined research line to later use other strategies to guide the search, such as Novelty Search [34], and being consistent with this previous work to make further comparisons. Therefore, we sampled the entire navigation environment to randomly select 100 initial conditions, each with a position and an orientation to come up with the configuration shown in Figure 5(b). This test set was held constant for all experiments; including the preliminary and exhaustive analysis.  Additionally, we took care that the training and test sets have no overlap."

We hope we answered correctly all your final questions and finally our manuscript can be accepted to be published in this special issue of the Mathematical and Computational Applications (MCA) journal part of the Multidisciplinary Digital Publishing Institute (MDPI).

Thanks a lot, sincerely,

The Authors